# Effect of Proton Pump Inhibitors on Risks of Upper and Lower Gastrointestinal Bleeding among Users of Low-Dose Aspirin: A Population-Based Observational Study

**DOI:** 10.3390/jcm9040928

**Published:** 2020-03-28

**Authors:** Luis A. García Rodríguez, Angel Lanas, Montse Soriano-Gabarró, Pareen Vora, Lucía Cea Soriano

**Affiliations:** 1Spanish Centre for Pharmacoepidemiologic Research (CEIFE), 28004 Madrid, Spain; luciaceife@gmail.com; 2Servicio de Aparato Digestivo, Hospital Clínico, University of Zaragoza, 50009 IIS Aragón, Zaragoza, Spain; angel.lanas@gmail.com; 3CIBERehd, Av. Monforte de Lemos 3–5, Pabellón 11, Planta 0, 28029 Madrid, Spain; 4Epidemiology, Bayer AG, 13353 Berlin, Germany; montse.soriano-gabarro@bayer.com (M.S.-G.) pareen.vora@bayer.com (P.V.); 5Department of Public Health and Maternal and Child Health, Faculty of Medicine, Complutense University of Madrid, 28040 Madrid, Spain

**Keywords:** aspirin, proton pump inhibitors, upper gastrointestinal bleeding, lower gastrointestinal bleeding

## Abstract

Estimates of the effect of proton pump inhibitors (PPIs) on risks of upper and lower gastrointestinal bleeding (UGIB and LGIB) among low-dose aspirin users in routine clinical practice are variable (UGIB) or lacking (LGIB). We aimed to establish these risks in the same observational study population. Using UK primary care data, we followed 199,049 new users of low-dose aspirin (75–300 mg/day) and matched non-users at start of follow-up to identify incident UGIB/LGIB cases. In nested case–control analyses, adjusted odds ratios (ORs) were calculated for concomitant PPI use vs. past (discontinued) PPI use among current low-dose aspirin users. For UGIB (*n* = 987), ORs (95% CIs) were 0.69 (0.54–0.88) for >1 month PPI use and 2.65 (1.62–4.3) for ≤1 month PPI use. Among the latter group, ORs (95% CIs) were 3.05 (1.75–5.33) for PPI initiation after start of aspirin therapy, and 1.66 (0.63–4.36) for PPI initiation on/before start of aspirin therapy. For LGIB (*n* = 1428), ORs (95% CIs) were 0.98 (0.81–1.17) for >1 month PPI use and 1.12 (0.73–1.71) for ≤1 month PPI use. Among low-dose aspirin users, maintaining PPI use (>1 month) was associated with a significantly reduced UGIB risk. Neither short nor long-term PPI use affected LGIB risk.

## 1. Introduction

Low-dose aspirin is widely prescribed for the secondary prevention of cardiovascular disease (CVD) as this benefit is well established in outweighing the risk of major bleeding [1]. The net value of low-dose aspirin among individuals without CVD is less certain because of concerns about the risks of major bleeding [1,2], the most common being that in the gastrointestinal tract. Decisions to prescribe low-dose aspirin to this patient population in clinical practice, therefore, requires a careful assessment of individual bleeding risks. Co-administration of an acid-suppressing agent in patients taking prophylactic low-dose aspirin has been shown to reduce the risk of aspirin-associated gastrointestinal bleeding [3,4,5,6,7]. Proton pump inhibitors (PPIs) are the preferred agents, being recommended in clinical guidelines for use in patients with gastrointestinal risk factors [8,9,10]. However, the gastroprotective effects of PPIs are limited to the upper gastrointestinal tract, with no protection against aspirin-induced bleeding in the lower gastrointestinal tract. Moreover, PPIs have been shown to affect the microbiota throughout the gastrointestinal tract, but in particular in the lower gastrointestinal tract [11], which could potentially increase susceptibility to enteric infections and gastrointestinal disorders [11,12,13,14]. Small clinical studies have shown that PPIs contribute to injury in the small bowel among low-dose aspirin users [15,16] yet whether this translates into a higher risk of lower gastrointestinal bleeding (LGIB) is unclear, as findings from the limited observational data on this topic have been mixed [5,17].

In population-based studies, estimates of the size of a PPI-induced reduction in upper gastrointestinal bleeding (UGIB) risk among users of low-dose aspirin have varied. Some studies have had limited power to produce precise estimates of effect, and definitions of drug use between studies have differed [5,6,7,18,19]. Further population-based studies are therefore required to quantify the effects that PPIs have on the risk of both UGIB and LGIB among these patients. Such studies require a robust study design to minimize threats to internal validity, such as confounding by indication and confounding arising from differences between users of low-dose aspirin/PPIs in characteristics and behaviors that are difficult to control for.

We used data from a population-based study of gastrointestinal bleeding set in UK primary care, using validated cases of both UGIB and LGIB. We used a matched cohort design with nested case–control analysis to quantify the effect that PPI use (vs. non-use) has on the risk of UGIB and LGIB among users of low-dose aspirin, using an appropriate reference group for PPI use that aimed to minimize confounding and selection bias. We analyzed both duration of PPI use (short- or long-term) and the timing of PPI initiation in relation to the start of low-dose aspirin therapy.

## 2. Materials and Methods

### 2.1. Study Design

We used The Health Improvement Network primary care database in the UK, which contains the electronic health records of approximately 6% of patients across the UK, who collectively are representative of the UK demographic [20,21]. This population-based database has been shown to be a valid data source for pharmaco-epidemiological research and for studying low-dose aspirin use in the UK – unrecorded use of over-the-counter obtained low-dose aspirin in the database has previously been shown to be minimal [22,23]. Details of our study design, including the study population inclusion/exclusion criteria, cohort matching, outcome identification and validation, identification and classification of co-variates, and sampling and frequency matching of controls in the nested case–control analyses (by age, sex and calendar year), have been published previously [24]. Briefly, between 1 January, 2000 and December 2012, 199,079 new users of low-dose aspirin and a 1:1 matched cohort of non-users of low-dose aspirin at the start of follow-up were followed for a maximum of 14 years (mean 5.4 years), and a total of 1843 incident cases of UGIB and 2763 incident cases of LGIB were identified. In addition to the traditional cohort matching criteria of age and sex, we applied an additional matching criterion–the number of primary care practitioner (PCP) visits in the year before start of follow-up. This served as a proxy measurement for general health with the aim of minimizing confounding from differences between users and non-users of low-dose aspirin at baseline. The index date was the date of the bleeding event for cases and a random date within their individual observation period for controls (5000 for UGIB and 10,000 for LGIB). In this present study, to evaluate the effect of PPI use vs. non-use on UGIB/LGIB risk among low-dose aspirin users, we retained only UGIB/LGIB cases and controls who were current users of low-dose aspirin (987 UGIB cases, 2160 UGIB controls; and 1428 LGIB cases, 4100 LGIB controls). All authors had access to the study data and reviewed and approved the final manuscript.

### 2.2. Low-Dose Aspirin and PPI Exposure

Use of low-dose aspirin among UGIB/LGIB cases and controls was categorized as follows: current use, when the supply of their most recent prescription lasted until (or over) the index date, or ended in the 30 days before the index date; past (discontinued) use, when low-dose aspirin use ended more than 1 month before the index date; never use, if there was no recorded low-dose aspirin prescription before the index date. Within current users of low-dose aspirin, PPI use was categorized using these same categories.

### 2.3. Statistical Analyses

Odds ratios (ORs) with 95% confidence intervals (CIs) were calculated using unconditional logistic regression adjusted for confounders using a stepwise forward approach. Patients with missing data on smoking, body mass index (BMI), alcohol consumption, estimated glomerular filtration rate or urban/rural residence were placed in a separate category ‘missing’ for that variable The fully adjusted model for the UGIB outcome was adjusted for the matching variables (sex, age and calendar year) and number of PCP visits in the year before the index date, alcohol consumption, smoking, history of UGIB, LGIB, unspecified gastrointestinal bleeding (GIB), peptic ulcer, pancreatic disease, polytherapy and use of clopidogrel, non-steroidal anti-inflammatory drugs (NSAIDs) and warfarin. The fully adjusted model for the LGIB outcome was adjusted for matching variables (sex, age and calendar year) and number of PCP visits in the year before the index date, alcohol consumption, smoking, body mass index, history of polyps, LGIB, unspecified GIB, peptic ulcer, gastro-esophageal reflux disease, inflammatory bowel disease, irritable bowel syndrome, polytherapy and use of clopidogrel, NSAIDs and warfarin. Polytherapy was defined as the number of different medications where a prescription lasted until/over the index date or lasted to within the 30 days before the index date. The main exposure of interest was current use of low-dose aspirin plus current use of a PPI (i.e., concomitant use) and, to minimize confounding from time-varying differences between PPI users and non-users that are difficult to control, the reference group was current use of low-dose aspirin plus past use of a PPI. To evaluate the likely effect of confounding by indication among individuals who had very recently started using a PPI (i.e., very short duration of use), we categorized duration of PPI use as either ≤1 month or >1 month. To explore possible effects relating to the timing of PPI initiation among low-dose aspirin users, we further categorized PPI use for patients with ≤1 month duration of use according to whether the PPI was prescribed after the start of low-dose aspirin therapy or on/before the start of low-dose aspirin therapy. An additional analysis was also undertaken to evaluate the effect of concomitant PPI use (of any duration vs. past PPI use) among current users of dual antiplatelet therapy (DAT; low-dose aspirin plus clopidogrel) on the risk of UGIB/LGIB. We did not perform any analyses for specific PPI agents. All analyses were conducted using STATA version 12.0.

## 3. Results

### 3.1. Characteristics of Cases and Controls

The main characteristics of UGIB/LGIB cases and controls are shown in Table 1; further characteristics can be found in Appendix A). Among the 987 incident cases of UGIB (all current users of low-dose aspirin), just over half (56.1%) were male and the mean age was 72.04 years (median 74 years). Among the 1428 incident cases of LGIB, approximately half were male (51.8%) and the mean age was 70.26 years (median 71 years).

### 3.2. Effect of PPIs on Risk of UGIB among Low-Dose Aspirin Users

The frequency distribution of low-dose aspirin and PPI use among UGIB cases and controls (all current users of low-dose aspirin) is shown in Table 2 along with associations between PPI use and UGIB risk. Among UGIB cases, 265 (26.8%) were concomitant users of a PPI. Among UGIB controls, 489 (22.6%) were concomitant users of a PPI. Compared with the reference group, concomitant current use of low-dose aspirin and a PPI was associated with a 31% reduced risk of UGIB when the PPI was used for more than 1 month (OR 0.69, 95% CI: 0.54–0.88) (with estimates similar in analyses stratified by primary/secondary CVD prevention population; see Appendix A for results of this post-hoc analysis), but with more than a two-fold increased risk during the first month of PPI use (OR 2.65, 95% CI: 1.62–4.33). When analyzing this further, there was no evidence of a significant increase in UGIB risk when the PPI therapy was initiated on or before the start of low-dose aspirin therapy (OR 1.66, 95% CI: 0.63–4.36), but when the PPI was prescribed after the start of low-dose aspirin therapy, the increase in risk during the first month of PPI treatment was evident (OR 3.05, 95% CI: 1.75–5.33) (Table 1). Among current users of DAT, the data were suggestive of a substantial reduction in UGIB risk (OR 0.62, 95% CI: 0.31–1.25) among patients who had used a PPI concomitantly; however, evaluations were limited by the small number of patients in the analyses (Appendix A).

### 3.3. Effect of PPIs on Risk of LGIB among Low-Dose Aspirin Users

The frequency distribution of PPI use among LGIB cases and controls (all current users of low-dose aspirin) is shown in Table 3 along with associations between PPI use and LGIB risk. Among LGIB cases, 444 (31.1%) were concomitant users of a PPI. Among LGIB controls, 934 (23.4%) were concomitant users of a PPI. Compared with the reference group, concomitant current use of low-dose aspirin and a PPI was not associated with a change in LGIB risk when PPI use was greater than 1 month in duration (OR 0.98, 95% CI: 0.81–1.17) or when PPI use was used for 1 month or less (OR 1.12, 95% CI: 0.73–1.71). Among the latter, the lack of association was still evident irrespective of when PPI use was initiated in relation to the start of low-dose aspirin therapy. Among current users of DAT, there was no evidence of a change in LGIB risk among patients who had used a PPI concomitantly (OR 0.97, 95% CI: 0.39–2.42) based on the small number of patients in this analysis (Appendix A).

Owing to the fact that the majority of LGIB cases among low-dose aspirin users in UK primary care are referred to a specialist but not hospitalized (25), we performed a post-hoc analysis in which we repeated the main LGIB analysis according to whether LGIB cases were only referred (less severe cases) or were hospitalized (more severe cases). Odds ratios did not differ between these two groups of cases, with minimal change seen from the main estimate: OR 0.97 (95% CI: 79–1.18) for only referred cases, and 1.04 (95% CI: 0.78–1.39) for hospitalized cases.

## 4. Discussion

In our large population-based study, we have shown that among patients using low-dose aspirin for CVD prophylaxis, those who continued with PPI co-therapy for at least 1 month had a significant 31% reduced risk of UGIB compared with those who discontinued PPI use (past users). However, co-therapy with a PPI, whether in the short-term or long-term, had no effect on their risk of LGIB. The increased UGIB risk seen in our study soon after the start of PPI co-therapy (i.e., among patients who had less than a month of PPI use) is most likely explained by protopathic bias due to the prescription of PPIs to treat prodromic symptoms of UGIB. Although interpreting the true effect of short-term PPI use is challenging in a non-randomized setting, results of our analyses separating out the timing of PPI initiation will help to elucidate the likely true effects of PPI co-prescription among low-dose aspirin users that are otherwise difficult to infer in observational studies when PPI use is grouped as single category.

Our study provides valuable data on the effect that PPIs have on LGIB risk in low-dose aspirin users because previous data on this topic have been limited and findings have been mixed [5,17]. In line with our results, a hospital-based case–control study by Nagata et al. [17] also found no association between PPIs and LGIB among users of low-dose aspirin in Japan based on 355 emergency hospitalizations for LGIB and 8221 non-bleeding control patients. A previous case–control study in Spain [5] found that among a combined group of users of antiplatelet agents, anti-inflammatory drugs or anticoagulants, patients using a PPI had a small increased risk of LGIB compared with those not using a PPI. This study, based on 415 hospitalized cases of LGIB, was underpowered to analyse effects of PPIs specifically among low-dose aspirin users. Our study had the advantage of including a much large number of LGIB cases and controls, providing greater power to detect significant differences between groups. We also did not restrict our study only to hospitalized cases but covered the whole spectrum of LGIB. This is an important factor in avoiding selection bias because we have previously shown the majority of LGIB cases among low-dose aspirin users in UK primary care (73%) are referred to a specialist but are not hospitalized [25].

Our findings, which suggest that PPIs are effective as protective agents against low-dose aspirin-related UGIB, support several findings from previous reports on the topic [5,6,7,18,19] yet further well-designed population-based studies are required to support or refute our findings. Clinical prescribers should consider prompt PPI co-prescription in these patients because our study also suggests that delaying PPI use to after the start of low-dose aspirin therapy (e.g., in response to when a patient presents with abdominal pain or dark blood in their stools) is associated with an increased risk of UGIB in the few weeks after initiating PPI treatment. This transient risk was not seen when PPI therapy was started at, or even before, the start of their aspirin therapy. Furthermore, the risk of gastrointestinal bleeding may be highest during the beginning of low-dose aspirin therapy, diminishing thereafter [26]. This highlights the importance of timely co-prescription of PPI therapy. It is not uncommon for patients to discontinue low-dose aspirin if they experience UGIB [27], thus timely PPI co-prescription could potentially help persistence with therapy. Previous research has shown that PPI co-prescription per se supports continuation of low-dose aspirin therapy [28], likely due to the protection against UGIB events and dyspepsia type symptoms. Persistence with low-dose aspirin therapy is important because its discontinuation has major clinical and economic consequences, being associated with a significantly increased risk of ischaemic CVD events in secondary prevention users [29]. Among UGIB/LGIB cases and controls in our study (all of whom were current low-dose aspirin users), less than a third were using a PPI concomitantly. Clinical guidelines currently advocate their use in patients with gastrointestinal risk factors [10,30], yet studies in the Netherlands [31] and Germany [32] suggest that even among this patient population they are under-prescribed. Reasons for this potential under-prescribing of PPIs are unknown but further research into this would be beneficial, especially as PPIs may have different effects in different patient subgroups. Older age is an important risk factor for UGIB [33], and there is evidence from the Oxford Vascular Study [34] that PPIs confer a greater absolute benefit in terms of UGIB protection among older individuals. Li et al. [34] estimated that among patients taking aspirin-based antiplatelet treatment for secondary prevention, the number needed to routinely treat with a PPI to prevent a disabling or fatal UGIB over 5 years was substantially lower among older individuals—338 for persons less than 65 years, and 25 for those aged 85 years or more. While it was beyond the scope of our study to evaluate the effect of PPIs among aspirin users of different age groups, we were able to evaluate the effect among those of another group at particular high risk of UGIB—users of DAT. Despite the limited power of the analyses and the resulting wide confidence intervals, our results for DAT suggest that the magnitude of effect might be similar to that seen among low-dose aspirin users—a 38% reduced the risk of UGIB, in line with a previous meta-analysis [35]. No association was seen between PPIs and LGIB risk among DAT users, as also found previously by Nagata et al. [17].

A key strength of our study is that we provide evidence of the risk of both UGIB and LGIB in relation to PPI use among low-dose aspirin users from the same study population and time period. Also, the large sample from a population representative of the UK demographic enabled a large number of cases and controls to be obtained. Another strength is the use of a suitable comparison group to evaluate the effect of current PPI use. Because PPI use is a marker of comorbidity per se [36], the use of past users of PPIs as the reference group (rather than non-users of PPIs) in the logistic regression analyses means that the effects seen in our study are less likely to be confounded than if we had used non-users of PPIs as the comparator. We also adjusted for confounders in the analyses, although we acknowledge that we cannot completely exclude the possibility of residual confounding. Additionally, some of the sub-group analyses had few exposed cases and controls in the different strata, resulting in estimates with wide CIs that need to be interpreted cautiously. Of note is that the frequency of short-term PPI use (a month or less duration) among UGIB/LGIB cases and controls was small, being even smaller in analyses partitioned by timing, and the confidence limits around the corresponding estimates were wide. Sample size considerations also limited further analyses according to gastrointestinal risk factors. Another limitation of our study is the possible misclassification of PPI use due to the availability of PPIs over-the-counter in the UK. Such misclassification will likely have been non-differential between cases and controls leading to a small underestimation of the observed effect in the UGIB analyses.

## 5. Conclusions

Our results indicate that patients using low-dose aspirin who are co-prescribed a PPI have a significant 31% reduced risk of UGIB compared with those patients who discontinue PPI therapy, and do not have an increased risk of LGIB. Timely PPI co-prescription affords greater protection against UGIB compared with PPI co-therapy that is delayed.

## Figures and Tables

**Table 1 jcm-09-00928-t001:** Characteristics of UGIB/LGIB cases and controls (all current users of low-dose aspirin).

	UGIB Cases *n* = 987		Controls *n* = 2160		LGIB Cases *n* = 1428		Controls *n* = 4100	
	*n*	%	*n*	%	*n*	%	*n*	%
**Sex**								
Male	554	56.1	1221	56.5	740	51.8	2116	51.6
Female	433	43.9	939	43.5	688	48.2	1984	48.4
**Age (years)**								
40–59	124	12.6	200	9.3	210	14.7	524	12.8
60–69	245	24.8	495	22.9	414	29.0	1270	31.0
70–79	367	37.2	837	38.8	523	36.6	1528	37.3
80–89	251	25.4	628	29.1	281	19.7	778	19.0
**Calendar year**								
2000–2004	237	24.0	508	23.5	282	19.7	812	19.8
2005–2010	488	49.4	1027	47.5	682	47.8	1957	47.7
2010 and beyond	262	26.5	625	28.9	464	32.5	1331	32.5
**Smoking**								
Non-smoker	351	35.6	850	39.4	562	39.4	1639	40.0
Current	158	16.0	256	11.9	160	11.2	542	13.2
Former	452	45.8	991	45.9	696	48.7	1844	45.0
Missing	26	2.6	63	2.9	10	0.7	75	1.8
**BMI (kg/m^2^)**								
15–19	51	5.2	71	3.3	53	3.7	118	2.9
20–24	213	21.6	511	23.7	310	21.7	958	23.4
25–29	366	37.1	840	38.9	580	40.6	1526	37.2
≥30	269	27.3	582	26.9	416	29.1	1263	30.8
Missing	88	8.9	156	7.2	69	4.8	235	5.7
**Alcohol (u/w)**								
None	218	22.1	400	18.5	272	19.0	829	20.2
1–9	435	44.1	1070	49.5	731	51.2	1955	47.7
10–20	131	13.3	311	14.4	185	13.0	623	15.2
21–41	62	6.3	94	4.4	78	5.5	202	4.9
≥42	15	1.5	18	0.8	25	1.8	38	0.9
Missing	126	12.8	267	12.4	137	9.6	453	11.0
**Polypharmacy**								
0–1	186	18.8	478	22.1	277	19.4	934	22.8
2–4	258	26.1	658	30.5	412	28.9	1229	30.0
≥5	543	55.0	1024	47.4	739	51.8	1937	47.2
**Townsend score**								
Deprived 1 (least deprived)	34	3.4	55	2.5	40	2.8	103	2.5
Deprived 2	217	22.0	525	24.3	367	25.7	994	24.2
Deprived 3	199	20.2	491	22.7	337	23.6	964	23.5
Deprived 4	198	20.1	440	20.4	286	20.0	833	20.3
Deprived 5 (most deprived)	194	19.7	403	18.7	228	16.0	725	17.7
Missing	145	14.7	246	11.4	170	11.9	481	11.7
**Other comorbidities**								
Myocardial infarction	137	13.9	300	13.9	205	14.4	555	13.5
IHD *	222	22.5	510	23.6	395	27.7	921	22.5
COPD	104	10.5	181	8.4	125	8.8	310	7.6
Hypertension	644	65.2	1482	68.6	947	66.3	2780	67.8
Hyperlipidemia	285	28.9	594	27.5	415	29.1	1241	30.3
Diabetes	267	27.1	563	26.1	308	21.6	1121	27.3
Peptic ulcer, uncomplicated/complicated	147	14.9	128	5.9	125	8.8	230	5.6
Peptic ulcer, uncomplicated	105	10.6	92	4.3	85	6.0	180	4.4
Peptic ulcer, complicated	64	6.5	46	2.1	54	3.8	67	1.6
IBD	12	1.2	20	0.9	68	4.8	59	1.4
IBS	72	7.3	95	4.4	150	10.5	229	5.6
Dyspepsia	304	30.8	505	23.4	456	31.9	998	24.3
Constipation	224	22.7	345	16.0	300	21.0	574	14.0
GERD	186	18.8	356	16.5	363	25.4	675	16.5
Prior UGIB	30	3.0	18	0.8	23	1.6	35	0.9
Prior LGIB	88	8.9	119	5.5	236	16.5	305	7.4
Prior GIB unspecified	24	2.4	20	0.9	23	1.6	31	0.8

* IHD does not include myocardial infarction. Note: Alcohol, BMI and smoking were ascertained any time before the index date using the most recent status/value as appropriate. Comorbidities were ascertained any time before the index date. Polypharmacy was taken as the number of different medications in the month before the index date. Note: There were 22 (1.5%) cases and 25 (0.6%) controls with a record of melena, and 76 cases (5.3%) and 96 controls (2.3%) with a record of polyps. Abbreviations: BMI, body mass index; COPD, chronic obstructive pulmonary disease; GERD, gastro-esophageal reflux disease; GIB, gastrointestinal bleeding; IBD, inflammatory bowel disease; IBS, irritable bowel syndrome; IHD, ischemic heart disease; LGIB, lower gastrointestinal bleeding; UGIB, upper gastrointestinal bleeding; u/w, units per week.

**Table 2 jcm-09-00928-t002:** ORs (95 CIs) for the association between concomitant use of low-dose aspirin and a PPI and the risk of UGIB.

	UGIB *n* = 987 *n* (%)	Controls *n* = 2160 *n* (%)	Crude OR (95% CI)	Adjusted OR (95% CI) *
**Current use of low-dose aspirin and past use of a PPI (reference group)**	223 (22.6)	423 (19.6)	1.0 (reference)	1.0 (reference)
**Current use of low-dose aspirin and current use of a PPI**	265 (26.8)	489 (22.6)	1.03 (0.82–1.28)	0.82 (0.65–1.04)
**Current use of low-dose aspirin and a PPI: ≤1 month PPI duration**	56 (5.7)	31 (1.4)	3.43 (2.15–5.47)	2.65 (1.62–4.33)
**Current use of low-dose aspirin and a PPI: ≤1 month PPI duration and prescribed after the start of low-dose aspirin**	46 (4.7)	22 (1.0)	3.97 (2.33–6.76)	3.05 (1.75–5.33)
**Current use of low-dose aspirin and a PPI: ≤1 month PPI duration and prescribed on/before the start of low-dose aspirin**	10 (1.0)	9 (0.4)	2.11 (0.84–5.20)	1.66 (0.63–4.36)
**Current use of low-dose aspirin and a PPI: >1 month PPI duration**	209 (21.2)	458 (21.2)	0.87 (0.69–1.09)	0.69 (0.54–0.88)
**Current use of low-dose aspirin and a PPI: >1 month PPI duration and prescribed after the start of low-dose aspirin**	89 (9.0)	222 (10.3)	0.76 (0.57–1.02)	0.63 (0.46–0.86)
**Current use of low-dose aspirin and a PPI: >1 month PPI duration and prescribed on/before the start of low-dose aspirin**	120 (12.2)	236 (10.9)	0.96 (0.73–1.27)	0.75 (0.56–1.00)
**Current use of low-dose aspirin and never use of a PPI**	499 (50.6)	1248 (57.8)	0.76 (0.63–0.92)	0.93 (0.76–1.14)

* Adjusted by matching variables (sex, age and calendar year) and number of PCP visits in the year before the index date, alcohol consumption, smoking, history of UGIB, LGIB, GIB, peptic ulcer, pancreatic disease, polytherapy and use of clopidogrel, NSAIDs and warfarin. Abbreviations: CI, confidence interval; GIB, gastrointestinal bleeding; LGIB, lower gastrointestinal bleeding; NSAID, non-steroidal anti-inflammatory drug; OR, odds ratio; PCP, primary care practitioner; PPI, proton pump inhibitor; UGIB, upper gastrointestinal bleeding.

**Table 3 jcm-09-00928-t003:** ORs (95 CIs) for the association between concomitant use of low-dose aspirin and a PPI and the risk of LGIB.

	LGIB *n* = 1428 *n* (%)	Controls *n* = 4100 *n* (%)	Crude OR (95% CI)	OR (95% CI) *
**Current use of low-dose aspirin and past use of a PPI (reference group)**	349 (24.4)	833 (20.3)	1.0 (reference)	1.0 (reference)
**Current use of low-dose aspirin and current use of a PPI**	444 (31.1)	961 (23.4)	1.10 (0.93–1.30)	0.98 (0.82–1.17)
**Current use of low-dose aspirin and a PPI: ≤1 month PPI duration**	38 (2.7)	73 (1.8)	1.24 (0.82–1.87)	1.12 (0.73–1.71)
**Current use of low-dose aspirin and a PPI: ≤1 month PPI duration and prescribed after the start of low-dose aspirin**	31 (2.2)	58 (1.4)	1.28 (0.34–0.47)	1.16 (0.73–1.84)
**Current use of low-dose aspirin and a PPI: ≤1 month PPI duration and prescribed on/before the start of low-dose aspirin therapy**	7 (0.5)	15 (0.4)	1.11 (0.45–2.76)	0.97 (0.38–2.53)
**Current use of low-dose aspirin and a PPI: >1 month PPI duration**	406 (28.4)	888 (21.7)	1.09 (0.92–1.30)	0.98 (0.81–1.17)
**Current use of low-dose aspirin and a PPI: >1 month PPI duration and prescribed after the start of low-dose aspirin**	192 (13.4)	389 (9.5)	1.18 (0.95–1.46)	1.05 (0.84–1.32)
**Current use of low-dose aspirin and a PPI: >1 month PPI duration and prescribed on/before the start of low-dose aspirin therapy**	214 (15.0)	499 (12.2)	1.02 (0.84–1.25)	0.91 (0.84–1.32)
**Current use of low-dose aspirin and never use of a PPI**	635 (44.5)	2306 (56.2)	0.66 (0.93–1.30)	0.82 (0.69–0.96)

* Adjusted by matching variables (sex, age and calendar year) and number of PCP visits in the year before the index date, alcohol consumption, smoking, BMI, history of polyps, LGIB, unspecified GIB, peptic ulcer, GERD, IBD, IBS, polytherapy and use of clopidogrel, NSAIDs and warfarin. Abbreviations: BMI, body mass index; CI, confidence interval; GERD, gastro-esophageal reflux disease; GIB, gastrointestinal bleeding; IBD, inflammatory bowel disease; IBS, irritable bowel syndrome; LGIB, lower gastrointestinal bleeding, NSAID, non-steroidal anti-inflammatory drug; OR, odds ratio; PCP, primary care practitioner; PPI, proton pump inhibitor; UGIB, upper gastrointestinal bleeding.

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
