# Peer review of "Effect of Proton Pump Inhibitors on Risks of Upper and Lower Gastrointestinal Bleeding among Users of Low-Dose Aspirin: A Population-Based Observational Study"

_jcm, 2020, doi:10.3390/jcm9040928_

Round 1
Reviewer 1 Report
The current research article Effect of proton pump inhibitors on risks of upper and lower gastrointestinal bleeding among users of low-dose aspirin: a population-based observational study - is a well designed which provides Estimates of the effect of proton pump inhibitors (PPIs) on risks of upper and lower gastrointestinal bleeding (UGIB and LGIB) among low-dose aspirin users in routine clinical practice are variable (UGIB) or lacking (LGIB).
Author Response
Reviewer 1
The current research article Effect of proton pump inhibitors on risks of upper and lower gastrointestinal bleeding among users of low-dose aspirin: a population-based observational study - is a well-designed which provides Estimates of the effect of proton pump inhibitors (PPIs) on risks of upper and lower gastrointestinal bleeding (UGIB and LGIB) among low-dose aspirin users in routine clinical practice are variable (UGIB) or lacking (LGIB).
Authors’ response: We thank the reviewer for these comments.

Reviewer 2 Report
Overall comments:
The authors analysed in a nested case-control study the effect of proton-pump inhibitors on the risk of upper and lower gastrointestinal bleeding in participants under low-dose aspirin therapy. The study design is adequate and the data was correctly analysed. Additionally, the research question is relevant. There are, though, some points of concern, as listed below. Furthermore, conflict of interest is present in this manuscript.
Major points:
Did you analyse, among the current users of aspirin, the duration of use (continuous) in relation to risk of UGIB/LGIB?
Statistical analyses: when having a small number of cases in a subgroup analyses and a substantial number of covariates included in the statistical model, you should be more careful in the way you are drawing conclusions. The rule of thumb of 1 covariate : 10 cases should be taken into consideration. Please it add to the discussion.
Minor points:
Results:
Baseline study population characteristics should be shown in the main file, not in the supplement. Tables 2 and 4 could go to the supplement.
Line 112: please remove first paragraph of this section.
Discussion:
Line 209: Please add reference.
Line 225: this sentence is way to strong to be done only based in the present findings. Please lower it down or remove it. This study suggests…. And results must be validated in other cohorts.
Materials and Methods:
Line 88: “All authord had access to …” this sentence should replaced to the “Authors Contributions”session
Line 99: For which confounders did you adjust? Should be explained also here.
Conclusions:
Please rephrase it, the way it is it sounds like the patients taking PPI had 1/3 the risk of non-takers.
Language: Moderate language review is necessary.
Reviewer 3 Report
Overall this is an important study with relevant clinical findings. However, edits need to be made before it can be accepted for publication.
Introduction:
-LGIB and UGIB abbreviations need to be spelled out before initial use (lines 50 and 52 respectively).
Methods:
- It would also be useful to calculate and include the crude odds ratios so that the reader can assess how adjusting for confounding impacted the study's results.
- The methods section needs to discuss the matching methods in writing as to which covariates were matched (second matching is only mentioned in table footer), as well as the methods for inclusion/exclusion of covariates (ie clinical significance, statistical significance, assessing colinearity, etc).
- Were sensitivity analyses conducted to assess if results varied based upon specific PPI agent? If not might want to mention that this wasn't assessed either in methods section or discussion
- Were sensitivity analyses conducted around potential impact of non PPI acid suppressant on study results such as H2RA? If not included in logistic regression or sensitivity analysis, this needs to be mentioned in the discussion as this may bias the results toward the null
- How was polypharmacy define (in logistic regression with clopidogrel, NSAIDS and warfarin)? Was it at least 1 overlap day (based upon fill date and day supply? Or was it receipt of Rx within a certain timeframe? This needs to be clarified
- Were sensitivity analyses conducted around potential impact of polypharmacy with rivaroxaban or dabagatran? Although this might be a small percentage of patients due to approval date of these agents being closer to end of study period, if not included in logistic regression or sensitivity analysis, this needs to be mentioned in the discussion as they would substantially increase risk of bleeding.
- In addition, Other commonly prescribed agents that increase risk of bleeding such as SNRIs or SSRIs were not assessed, and should at least be mentioned as a limitation
Results
-The text after Title "3. Results" and before "3.1 Characteristics of cases and controls" appears to be wording from authorship guidelines and should be removed.
-Supplementary Tables 1 and 2 should be combined and included in the main text of the article so that the reader can assess similarities and differences of the populations of cases and controls.
-Title "3.3" says "UGIB" when it should read "LGIB"
Discussion
-Sentence starting on line 235 is overstating the results of the study. This study did not assess of compare results based upon clinical adherence.
Round 2
Reviewer 2 Report
The authors increased significantly the quality of the manuscript. Some details still need to be corrected though.
Line 124: The reason why you did not analyze types of PPI should be explained.
Tables 1 and 2 should be merged into one table (Table 1). Also, please select the relevant information to be shown in this table, and shift secondary information to the supplementary material, e.g. “other comorbidities”.
Please explain in the “Statistical Analyses” section how you dealt with missing data.
Line 280: “support” instead of “supports. Please review the English language from the whole manuscript.
Author Response
Comment 1: Line 124: The reason why you did not analyze types of PPI should be explained.
Authors’ response: The main reason we did not analyse different types of PPIs was that we are unaware of any scientific/biological rationale as to a potential differential effect and therefore a reason for investigating this. Furthermore, such analysis would be limited by the sample size, with the estimates produced being very imprecise.
Comment 2: Tables 1 and 2 should be merged into one table (Table 1). Also, please select the relevant information to be shown in this table, and shift secondary information to the supplementary material, e.g. “other comorbidities”.
Authors’ response: We have now combined Tables 1 and 2 into a single table (Table 1) and renamed the subsequent tables accordingly. We have removed secondary information into a new Supplementary table (Supplementary Table 1) and have renamed all subsequent Supplementary material accordingly.
Comment 3: Please explain in the “Statistical Analyses” section how you dealt with missing data.
Authors’ response: We have now added the following text to the ‘Statistical Analyses’ section of the methods, starting on line 102:
“Patients with missing data on smoking, BMI, alcohol consumption, estimated glomerular filtration rate or urban/rural residence were placed in a separate category ‘missing’ for that variable.”
Comment 4: Line 280: “support” instead of “supports.
Authors’ response: We thank the reviewer for noticing this and have changed the word ‘supports’ to ‘support’. We have also reviewed the English language of the whole manuscript as recommended by the reviewer.
Reviewer 3 Report
The revisions the authors made helped strengthen the quality of this study. I believe their responses were adequate and recommend the acceptance of this manuscript for publication.
Author Response
We thank the reviewer for these comments.